# Mass Bleaching in Intertidal Canopy-Forming Seaweeds after Unusually Low Winter Air Temperatures in Atlantic Canada

**Ricardo A. Scrosati ***  and **Nicole M. Cameron**

Department of Biology, St. Francis Xavier University, Antigonish, NS B2G 2W5, Canada; ncameron@stfx.ca
* Correspondence: rscrosat@stfx.ca

**Abstract:** Contemporary climate change is increasing the occurrence of extreme weather events, heat waves being common examples. Here, we present visual evidence of mass bleaching in intertidal seaweeds following an unusually severe cold snap in Atlantic Canada. In February 2023, the air temperature on the Nova Scotia coast dropped below −20 °C for the first time in at least ten years. Such extreme temperatures lasted for several hours at low tide and were followed by extensive bleaching in whole thalli of the canopy-forming algae *Chondrus crispus* and *Corallina officinalis*. The loss of these foundation species might negatively impact intertidal biodiversity.

**Keywords:** cold spell; intertidal; seaweed

Foundation species are spatially dominant species that increase habitat complexity with their body structures and limit local abiotic stress. Thanks to those properties, dense stands of foundation species often host a high diversity of small species. Examples of such organisms are trees, shrubs, and seaweeds [1,2]. In particular, canopy-forming seaweeds are common in rocky intertidal habitats, where they often increase local biodiversity through the provision of shelter from various stresses [3,4]. Therefore, intertidal macroalgal beds are vital for coastal biodiversity preservation and ecosystem functioning.

Intertidal habitats experience underwater conditions at high tide and aerial conditions at low tide every day; therefore, extreme weather events can greatly affect the survival of intertidal organisms. This concept is becoming increasingly relevant because weather extremes are intensifying with the ongoing climate change [5,6]. A dramatic example occurred recently on the NE Pacific coast. In the summer of 2021, a severe heat wave broke historical records of maximum air temperature and caused a mass mortality of intertidal organisms on that coast [7].

Although current climate change eminently represents warming on a global scale, unusually severe cold stress in winter can take place at regional scales. For example, the winter temperature gradient between the Arctic and middle latitudes is weakening because Arctic winter temperatures are rising more quickly. This phenomenon is known as Arctic amplification, and can lead to polar air being transported to middle latitudes in winter, resulting in cold air outbreaks in Eurasia and North America [8,9]. In early February 2023, a period of unusually cold weather took place in Atlantic Canada [10]. Although the extreme conditions did not last for more than two days, air temperatures were markedly negative and disrupted human activities in many ways [11,12]. This cold snap was followed by dramatic effects on intertidal macroalgal stands as well.

For instance, the red algae *Chondrus crispus* Stackhouse (Gigartinaceae) and *Corallina officinalis* Linnaeus (Corallinaceae) often form extensive beds in rocky intertidal habitats in southeastern Nova Scotia [13]. Both species are known to host various invertebrate species [14,15]; as such, they are prominent foundation species. Our field surveys performed on the 15th of April 2023 at a typical location on this coast, Western Head (N 43.9896, W 64.6607), revealed extensive bleaching in both algae (Figure 1) in a way not seen in at least the previous ten years [16]. At two other locations situated north (Duck Reef, N 44.4913,

W 63.5270) and south (West Point, N 43.6533, W 65.1309) of Western Head, spanning a linear distance of 160 km, algal bleaching was also evident. Algal bleaching entails the loss of photosynthetic pigments and represents disruptive stress due to the inability of bleached tissues to recover, which leads to mortality [17,18]. In intertidal Gigartinaceae specifically, bleaching is common in summer as a result of intense heat, irradiance, and desiccation during low tides. Such seasonal bleaching progresses from a deep red color of algal fronds in winter, to a yellow-green color in spring, to full loss of color in summer, which mainly occurs in distal frond areas that are not protected by frond crowding through self-shading [19]. In the case hereby described for *C. crispus*, however, the color progression proceeded from typical deep red in early winter directly to the bleaching of whole thalli shortly after the February cold snap (Figure 1).

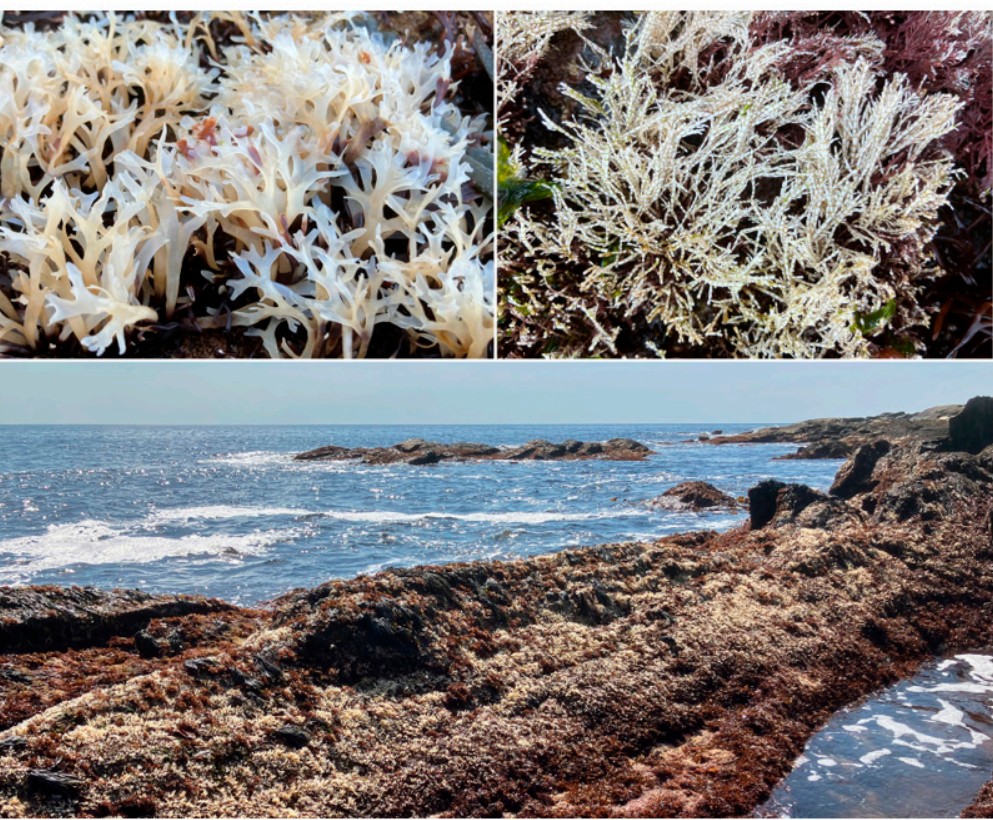

**Figure 1.** Whole-thallus bleaching in *Chondrus crispus* (**top left**) and *Corallina officinalis* (**top right**), and extensive bleaching in intertidal beds of these foundation species (**bottom**) photographed by R.A.S. at low tide on the 15th of April 2023 at Western Head, on the open Atlantic coast of Nova Scotia, Canada.

At a coastal weather station (Western Head Station, N 43.99, W 64.6642) near the intertidal habitats that we surveyed at Western Head, air temperature between the 3rd and 4th of February 2023 dropped below −20 °C for the first time in the last ten years [10]. These extreme conditions lasted for several hours and included the period of low tide in the early hours of the 4th of February. It is therefore likely that severe cold stress may have occurred at low tide during that time (see a possible example for intertidal corals in [20]). An additional factor contributing to this widespread bleaching event may have been the mild thermal conditions in December 2022 and January 2023 relative to the previous ten winter seasons [10], which may have not allowed algal thalli to acclimate enough to intense cold stress before the February cold snap struck. Overall, this case of mass bleaching in intertidal seaweeds after extreme winter air temperatures adds to the more numerous examples of bleaching documented for subtidal organisms (mainly corals) in relation to extreme water temperatures often during marine heat waves [21–24].

While the 2021 NE Pacific heat wave has been attributed to climate change [25], an equivalent conclusion for the 2023 NW Atlantic cold snap is missing, although it would not be surprising given that cold spells in eastern North America are thought to be favored by climate change [9]. Overall, the extensive bleaching experienced by macroalgal beds may have deprived intertidal environments from prominent foundation species that were able to host many small species. Whether these stands can regain their historical abundance is unknown because this depends, among other factors, on the future occurrence of winter cold spells (and probably summer heat waves as well). As extreme weather events are expected to increase in frequency with climate change [6,26], the biodiversity of these coastal ecosystems may experience significant changes in the future.

**Author Contributions:** R.A.S. and N.M.C. conceived the study and surveyed the coast, R.A.S. wrote the manuscript, and N.M.C. provided editorial comments. Both authors have read and agreed to the published version of the manuscript.

**Funding:** This study was funded by a Discovery Grant (#311624) awarded by the Natural Sciences and Engineering Research Council of Canada (NSERC) to R.A.S.

**Institutional Review Board Statement:** Not applicable, as this study did not involve humans or animals. This field study required no field permits or ethical review.

**Data Availability Statement:** Not applicable.

**Conflicts of Interest:** The authors declare no conflict of interest.

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
