# Peer review of "Mass Bleaching in Intertidal Canopy-Forming Seaweeds after Unusually Low Winter Air Temperatures in Atlantic Canada"

_diversity, doi:10.3390/d15060750_

Round 1

Reviewer 1 Report

This research manuscript is well structured, focuses on key themes, and provides interesting images for macroalgae research. No comments and critiques are necessary for the research scope and content. Therefore, publication of the paper is recommended.

Author Response

Thank you for recommending the publication of our manuscript.

Reviewer 2 Report

Please add one or several paragraphs considering bleaching on the Great Barrier Reef. Is it the same story on your environment? What is different? Can your environment better recuperate? --- There is plenty of literature to help you to look at the greater picture.

Author Response

Please note that the modifications that we have applied to the manuscript are highlighted in yellow.

There is a fundamental difference between our study system and the Great Barrier Reef. Our article focuses explicitly on extreme weather events caused by extreme air temperatures, as the studied seaweeds are intertidal and, thus, experience air temperatures during low tides. In contrast, the organisms that are experiencing bleaching in the Great Barrier Reef (corals) are subtidal, so they are never affected directly by air temperatures (instead, their bleaching is caused by extreme water temperatures). Therefore, adding one or more paragraphs exclusively to discuss bleaching in those corals will most likely divert the readers’ attention away from the main focus of our article. Instead, we have added a brief text including references on Great Barrier Reef corals at the end of the previous-last paragraph. We also note that the manuscript provides general information on what constitutes normal bleaching in intertidal seaweeds, which readers will be able to use to contrast with our unusual findings. In addition, we found a recent article on intertidal corals experiencing bleaching also due to cold stress at low tide (Rich et al. 2022), which we are citing in the revised manuscript to broaden its context.

Regarding the chances of recovery of our study system after the observed mass bleaching, we provide a discussion in the last paragraph of the manuscript. As this extreme bleaching event has not been reported for this cold-temperate intertidal system before, nobody really knows how (or if) it will recover, especially given the more frequent extreme weather events predicted because of climate change. Our intention is to monitor this system for years to come.

We remain grateful to this reviewer for the constructive comments.

Reviewer 3 Report

This is an important note that underlined the current and expected detrimental effects of on-going climate changes on a global scale. Recommend to publish following some recommendations, as follows:

Similar extreme, short whether events, whether they are heat waves or cold snaps could be mentioned in the text. One example is in the Mediterranean Sea, particularly in the Levantine basin; Israel, Egypt, Cyprus, where heat waves from few hours to several days have been common for decades. One question is whether these bleaching events really destroys the bed-macroalgae in the intertidal, or are these seaweeds able to recover shortly after, or following several weeks in the same season?

Some quantitative evaluation of the extent of the damage could be of interest.

Author Response

We thank this reviewer for the overall support.  Please note that the modifications that we have applied to the manuscript are highlighted in yellow.

We have added references about bleaching in subtidal organisms following extreme water temperatures at the end of the previous-last paragraph. On the possible recovery of the studied seaweeds after their mass bleaching, we provide a discussion in the last paragraph of the manuscript. As this extreme event has not been reported for this cold-temperate intertidal system before, nobody really knows how (or if) it will recover, especially given the more frequent extreme weather events predicted because of climate change. Our intention is to monitor this system for years to come.

Regarding the spatial extent of the bleaching event, our focal coastal location was Western Head, but we also visually surveyed two other locations (Duck Reef and West Point) that span a linear distance of 160 km, where we also found clear evidence of algal bleaching. We now report this in the revised manuscript.

We appreciate the constructive comments from this reviewer.

Round 2

Reviewer 2 Report

good revision